# Improving Access to Radiotherapy in Gauteng: A Framework for Equitable Cancer Care

**DOI:** 10.3390/ijerph22071071

**Published:** 2025-07-03

**Authors:** Portia N. Ramashia, Pauline B. Nkosi, Thokozani P. Mbonane

**Affiliations:** 1Department of Environmental Health, Faculty of Health Sciences, University of Johannesburg, Johannesburg 2000, South Africa; tmbonane@uj.ac.za; 2Department of Medical Imaging and Radiation Sciences, Faculty of Health Sciences, University of Johannesburg, Johannesburg 2000, South Africa; 3Faculty of Health Sciences, Durban University of Technology, Durban 4001, South Africa; paulinen1@dut.ac.za

**Keywords:** access, cancer care, framework, health equity, health policy, healthcare system, public health, radiotherapy, South Africa, sustainable development goal

## Abstract

Radiotherapy, a critical component of cancer treatment, faces significant challenges in Gauteng, South Africa. These disparities hinder the achievement of Sustainable Development Goal 3, primarily due to systemic issues, socioeconomic barriers, and limitations within the health system. This article presents the House framework, designed to enhance access to radiotherapy services by integrating the WHO Health Systems framework with the dimensions of access proposed by Penchansky and Thomas. The framework is visually represented as a house, with Policy & Governance as the foundation, WHO building blocks as pillars, and Equitable Cancer Care and Improved Outcomes as the roof. A mixed-methods approach was utilized, combining quantitative data from radiotherapy facilities and qualitative insights from healthcare professionals to identify barriers and potential solutions. Findings indicate significant disparities in resource distribution and accessibility between public and private institutions, compounded by socioeconomic factors like transport costs and lack of awareness. The article discusses innovative proposed framework using the 5As of access as potential solutions. The House framework serves as a valuable tool for policymakers and healthcare providers aiming to improve radiotherapy access and promote equitable cancer care in Gauteng, ultimately working towards reducing disparities in cancer outcomes.

## 1. Introduction

Radiotherapy is an indispensable component of cancer management, offering effective treatment for a wide range of malignancies. However, access to this essential modality remains unevenly distributed worldwide, leading to significant disparities in treatment outcomes [1,2,3,4]. Globally, high-income countries have significantly more radiotherapy facilities and trained personnel per capita than low- and middle-income countries, resulting in substantial differences in cancer mortality rates [5,6,7]. These global inequities are mirrored in South Africa, a country with a complex healthcare landscape [5]. South Africa presents a unique paradox, possessing pockets of advanced medical infrastructure while simultaneously grappling with a high ratio of patients to radiotherapy machines, particularly in the public sector [5].

Socioeconomic factors, such as poverty and geographical barriers, further exacerbate these disparities, impacting patients’ ability to access timely and appropriate cancer care [5]. Patients with limited financial means face challenges in affording transportation, accommodation, and other associated costs. Such deficits diminish the potential to cure cancer, control its spread, and ease suffering, all benefits that radiotherapy can provide to over half of cancer patients [8,9,10]. These challenges are particularly acute in Gauteng, South Africa’s most populous province. Escalating cancer incidence, coupled with constraints in equipment availability, human resources, and training, creates a particularly challenging environment for radiotherapy access in Gauteng [4,7]. To address these challenges, this study introduces the House framework, a novel, context-specific approach integrating the WHO Health Systems framework and the 5As of access, to provide a comprehensive understanding of factors impacting radiotherapy access within Gauteng’s unique healthcare landscape.

This paper is part of a larger project aimed at developing a framework for equitable cancer care in Gauteng. This framework is informed by findings from previous phases of the project, which employed mixed-methods approaches to provide a comprehensive understanding of the challenges [11,12]. These phases include an analysis of patient experiences of socioeconomic and demographic challenges to accessing radiotherapy, an exploration of structural quality indicators, and a quantitative analysis of key time intervals from diagnosis to treatment. This framework seeks to address the critical gap in equitable access to radiotherapy in Gauteng, focusing on infrastructure optimization, workforce development and retention, and strategic resource allocation. Therefore, this paper aims to present the developed framework to improve access to radiotherapy in Gauteng. By doing so, it seeks to contribute to improved cancer care outcomes and reduced disparities in the region.

## 2. Materials and Methods

### 2.1. Study Design

This project employed a concurrent mixed-methods design to develop a framework for improving access to radiotherapy services in Gauteng, South Africa. The study comprised five phases: a systematic review of barriers to radiotherapy access; a quantitative analysis of key time intervals from diagnosis to treatment; a mixed-methods assessment of structural quality indicators at radiotherapy facilities; a qualitative study exploring patient experiences; and framework development integrating findings from the previous phases.

### 2.2. Ethical Considerations

The project received ethical approval from the Faculty of Health Sciences Research Ethics Committee at the University of Johannesburg (REC-2509-2023). Gatekeeper permission was obtained from the Gauteng Health (NHRD ref no.: GP_202311_078), and all participating private oncology practices in Gauteng province. The project was conducted in accordance with the ethical principles outlined in the Declaration of Helsinki and in compliance with South Africa’s national research ethics guidelines. Written informed consent was obtained from all participants prior to data collection, after providing them with detailed information about the study’s purpose, procedures, risks, and benefits. Participants were informed of their right to withdraw from the study at any time without consequence.

### 2.3. Methods

The project, aimed at developing a framework for equitable cancer care, comprised five phases, each designed to address specific aspects of radiotherapy access in Gauteng.

#### Phase 5: Framework Development

This phase of the project was built upon the findings of several previous research activities. First, a systematic review: Barriers to Radiotherapy Access in Sub-Saharan Africa for Patients with Cancer: A Systematic Review was conducted to identify barriers to radiotherapy access in sub-Saharan Africa. Databases searched included PubMed, Scopus, Web of Science, and African Journals Online [13]. The review included 91 studies. Table 1 below shows the search terms that were used in combination with Boolean operators “AND”, “OR” and “Not”, “Breast cancer”; “Breast carcinoma”; “Breast neoplasm”; “Breast Tumor”; “Factors”; “Determinants”; “Barriers”; “Challenges”; “Delayed treatment”; “Time-to-Treatment”; “Provider delay”; radiotherapy delay”; “Treatment delay”; “Health system delay”; “Healthcare delivery”; “healthcare access”; “health service accessibility”; “Africa”; “ sub-Saharan Africa”; “low-middle income” and the names of each of the 48 sub-Saharan African countries. The systematic review identified key barriers to radiotherapy access in sub-Saharan Africa, including long distances to facilities, limited resources, and a shortage of trained personnel. These findings highlighted the need for the framework to address health system-level challenges and patient-related barriers. The review also emphasized the impact of socio-cultural beliefs, and the stigma associated with cancer on treatment-seeking behavior, informing the framework’s focus on community-based education and culturally sensitive communication strategies [13]. 

This was followed by the quantitative analysis of time intervals: Improving Access to Radiotherapy Services in Gauteng: Improving Access to Radiotherapy Services in Gauteng. A retrospective quantitative analysis was conducted to measure time intervals from diagnosis to treatment at radiotherapy facilities in Gauteng. Data were collected from 800 patient files (400 from each of two public radiation oncology centers). Statistical methods were used to calculate and analyze time intervals (e.g., means, medians, ranges) [14]. The quantitative assessment revealed systemic delays in the radiotherapy pathway in Gauteng, with significant time intervals between diagnosis and treatment initiation. The mean time for the first consultation was 8.32 months, and the average time to CT simulation was 13.38 months. These delays underscored the need for optimizing referral processes and addressing resource constraints within the framework. Then, we undertook the Mixed-Methods Assessment of Structural Quality Indicators: Improving Access to Radiotherapy: Exploring Structural Quality Indicators for Radiotherapy in Gauteng Province, South Africa. A concurrent triangulation mixed-methods design was employed to explore structural quality indicators of radiotherapy services in Gauteng province and their impact on patient access. Quantitative data on structural quality indicators were collected from two public and eleven private facilities. Qualitative data were collected through semi-structured interviews with eight heads of radiation oncology departments, radiation oncologists, medical physicists, and radiation therapists. The quantitative and qualitative data were integrated to provide a comprehensive understanding of the research problem [15]. The mixed-methods assessment highlighted disparities in structural quality indicators between public and private radiotherapy facilities in Gauteng. Public facilities generally had a higher patient load with limited equipment and staffing compared to private centers. The qualitative data from interviews with healthcare professionals revealed challenges in maintaining equipment in public facilities due to budget constraints and bureaucratic processes.

Finally, we produced the Qualitative Study of Patient Experiences: Experiences of Cancer Patients: Socio-economic and Demographic Challenges to Radiotherapy Access in Gauteng, South Africa. A descriptive cross-sectional, qualitative design using in-depth interviews was conducted to explore patient experiences and socioeconomic and demographic challenges to accessing radiotherapy services in Gauteng Province, South Africa. Face-to-face interviews were conducted with 25 patients (Participants demographic information-Appendix A) over 2 months. Participants were recruited using convenience sampling. The interview guide was developed based on the 5 A’s framework. Thematic analysis was used, and the Atlas.ti software version 24 facilitated data organization, coding, and theme development. Inter-coder reliability was assessed to ensure rigor and trustworthiness. The credibility of the qualitative data was enhanced through member checking and peer debriefing. The Consolidated Criteria for Reporting Qualitative Research checklist guided the reporting of the study’s findings. The qualitative study of patient experiences revealed key themes such as “Financial Burden” and “Lack of Information”. Patients reported significant out-of-pocket expenses for transportation, accommodation, and meals, with some having to sell their belongings to pay for transport to the hospital.

To ensure the trustworthiness of the qualitative data, inter-coder reliability was established by having an independent coder analyze a subset of transcripts, with discrepancies resolved through discussion and consensus [16]. The researchers also maintained reflexivity throughout the study, documenting their biases and assumptions to ensure findings were grounded in the data. The quantitative data, on the other hand, were analyzed with the assistance of a statistician from the University Statistical Consultation Service using SPSS version 29. Data were cleaned and verified for accuracy prior to analysis. The statistician advised on the selection of appropriate statistical tests, including t-tests, chi-square tests, and regression analysis, to examine relationships between key variables.

The House framework visually and conceptually integrates findings from the systematic review, quantitative assessment, mixed methods assessment, and qualitative study. The findings from each phase of the study were systematically integrated. The systematic review identified key constructs for inclusion. The quantitative data highlighted disparities that the framework needed to address. The qualitative data provided rich contextual understanding, ensuring the framework was grounded in oncology healthcare professional’s perspective and the patient experiences. The House metaphor illustrates how different components work together, with the foundation representing ‘Policy & Governance,’ the pillars symbolizing the core WHO building blocks, and the roof signifying ‘Equitable Cancer Care & Improved Outcomes’ [17]. This framework combines the strengths of the WHO Health Systems framework and Penchansky and Thomas’s 5As of access to comprehensively address the challenges in Gauteng [18]. This holistic approach informs the identification of targeted interventions, as evidenced by the qualitative study of patient experiences [19].

The study employed concurrent mixed methods design to ensure comprehensive data triangulation; the data collection process is visually represented in Figure 1 [20]. Quantitative data on structural quality indicators were combined with qualitative data from interviews to provide a multi-faceted understanding of radiotherapy access. Data were collected concurrently to identify emerging themes. Integration during analysis allowed for nuanced insights, such as comparing delays in starting radiotherapy with qualitative accounts of the patients’ challenges [21]. While the focus on Gauteng Province and sample size warrants caution in generalizing findings, the design enhanced data triangulation, providing a robust understanding of radiotherapy access.

## 3. Results

The House framework visually represents the integrated framework developed in this study to improve access to radiotherapy services in Gauteng, South Africa. This framework combines the strengths of the WHO Health Systems framework and Penchansky and Thomas’s 5As of access Figure 2. To illustrate the integration of findings from the different phases of this study, Table 1 presents a synthesis of the main themes identified in the different phases.

### 3.1. Foundation: Policy & Governance

As the foundation of the House framework, the Policy & Governance element is fundamental to establishing a supportive environment for cancer care by directly influencing the 5As of access, which effective policy and governance should ensure as follows: Availability of adequate radiotherapy resources, improve Accessibility by addressing geographical and structural barriers, enhance Affordability through financial support mechanisms, promote Acceptability through culturally sensitive care, and ensure appropriate Accommodation of patient needs within the healthcare system. This section examines the current challenges in policy and governance related to cancer control planning and resource allocation in Gauteng. These challenges often manifest as systemic factors such as policy gaps, regulatory barriers, and governance issues, which pose significant challenges to the delivery of radiotherapy services.

#### 3.1.1. The Challenges

Lack of effective cancer control planning, resource allocation, and leadership at the national and provincial levels.Inadequate cancer planning as depicted in the literature on cancer control in South Africa.

#### 3.1.2. The Impact of Inadequate Policy and Governance

Resource disparities include the uneven distribution of resources, leading to shortages in public facilities and compromised quality of care. This impacts the availability of services; public facilities generally had a higher patient load with limited equipment and staffing compared to private centers.

Equipment maintenance issues: delays in equipment maintenance due to budget constraints and bureaucratic processes also impacts availability. As one healthcare professional mentioned in the structural quality assessment phase, “We often have to wait months for repairs, which leads to treatment delays.”Financial burden on patients: lack of financial assistance programs and inadequate health insurance coverage, resulting in significant out-of-pocket expenses for patients. This directly affects Affordability. As stated by the one patient in the patients’ experience phase, “I had to sell my belongings to pay for transport to the hospital. It was a big financial strain on my family.”Information gaps: insufficient patient education and support, leading to confusion about the treatment process and a desire for more information from healthcare providers. This impacts Accommodation and Acceptability. As one patient noted in the patients’ experience phase, “I didn’t really understand what radiotherapy was or what to expect. I felt like I was just going through the motions”.

#### 3.1.3. The Proposed Solutions: A Multi-Pronged Approach

Addressing the challenges in policy and governance requires a multi-faceted approach that aligns with the 5As of access and leverages the potential of AI. The following proposed solutions aim to create a more enabling environment for equitable cancer care.

##### Develop Comprehensive National Cancer Control Plans

In Gauteng, the absence of a fully implemented comprehensive cancer control plan has been identified as a barrier to equitable cancer care. While South Africa has a National Cancer Strategic Framework, its effective implementation in Gauteng faces challenges [22]. This is reflected in the delays experienced by patients accessing radiotherapy services. Specifically, the systematic review identified that the lack of coordination between different levels of care contributes to these delays. Nationally, the absence of clear guidelines and standardized protocols for referral pathways contributes to fragmented care and inefficiencies, directly impacting the Accessibility and Accommodation aspects of the 5As framework. To address this, a national cancer control plan should address the entire cancer continuum from prevention to palliative care. This plan should include specific, measurable, achievable, relevant, and time-bound objectives. Furthermore, this national cancer control plan should address the shortage of oncology healthcare staff, including radiation oncologists, medical physicists, radiation therapists, and oncology nurses, particularly in public facilities, as this contributes to disparities in access and quality of care. Strategic investment in radiotherapy infrastructure and technology is also essential to meet the growing demand for cancer treatment in Gauteng. Many countries have demonstrated the positive impact of well-designed and implemented national cancer control plans. For example, countries with a national cancer control plan are more likely to have comprehensive, coherent, or consistent plans [23,24]. These plans need to address the entire cancer care continuum, from prevention and early detection to treatment and palliative care.

##### Increase Investment in Cancer Care

To address the challenges in Gauteng’s public facilities, an increase in investment in cancer care is crucial. This can be achieved by allocating a significant percentage of the provincial health budget to cancer services, with a prioritized focus on radiotherapy infrastructure and workforce development, and retention in public facilities. This allocation should be phased in over time, with clear, measurable targets for each year. Public facilities in Gauteng face capacity constraints due to limited equipment and staffing. The disparities between public and private facilities show that public facilities serve a larger patient population with fewer resources. Increased investment is crucial to address these disparities and improve access to timely, quality radiotherapy services [2,10,25,26,27]. Modernizing equipment and ensuring maintenance should also be considered [28]. To manage and oversee the allocation of these resources, a dedicated cancer fund can be established within the provincial health department. A transparent procurement process should be developed to ensure that funds are used efficiently and effectively. Key indicators such as the number of functional radiotherapy machines in public facilities, the number of trained radiation oncologists and therapists, and waiting times for radiotherapy treatment should be tracked and published regularly to ensure accountability and transparency [10,29]. Firstly, a compelling ethical argument must be presented, emphasizing that access to quality cancer care, including radiotherapy, is a fundamental human right and that failure to invest adequately leads to preventable suffering. Secondly, the economic benefits of investing in cancer treatment should be highlighted, demonstrating its cost-effectiveness compared to managing advanced-stage disease and its contribution to a healthier, more productive workforce [30]. To maximize impact, opportunities should be explored for reallocating funds within the existing provincial health budget by improving management and reducing waste.

Potential funding sources include allocating a dedicated portion of the provincial health budget to cancer care, seeking conditional grants from the national government, forging public–private partnerships, pursuing international funding opportunities and establishing a dedicated cancer fund within the provincial health department [31].

##### Promote Collaboration Between Public and Private Sectors

This impacts Accessibility and Availability. Collaboration between public and private healthcare providers should be encouraged to improve access to cancer care services. This can involve sharing resources, coordinating treatment pathways, and developing joint initiatives to address specific challenges. To enhance access to cancer care services, a formal Gauteng Cancer Care Collaborative (GCCC) should be established [32,33,34]. This joint task force should be comprised of representatives from the provincial health department, public hospitals, private oncology centers, academic institutions, and patient advocacy groups. There is a need for a holistic approach to cancer care, and the GCCC can facilitate this by fostering communication and collaboration across different stakeholders. Collaboration between public and private sectors can improve access to cancer care services by sharing resources and coordinating treatment pathways [34,35]. The recently passed National Health Insurance (NHI) policy in South Africa provides a framework for this collaboration, aiming to ensure that all citizens have access to quality health services, including cancer care, regardless of their socioeconomic status. Under the NHI, the Gauteng Cancer Care Collaborative can play a crucial role in aligning public and private sector efforts to deliver comprehensive cancer care. This joint task force should be comprised of representatives from the provincial health department, public hospitals, private oncology centers, academic institutions, and patient advocacy groups. By fostering communication and collaboration across different stakeholders, the GCCC can facilitate a holistic approach to cancer care.

The NHI can incentivize private sector participation by creating clear contracting mechanisms and service-level agreements that ensure fair compensation for services rendered [31,36]. This can help to mitigate the risk of “cream skimming,” where private providers selectively offer high-profit services, by ensuring that they are also incentivized to provide care for complex and less profitable cases. The NHI may ensure equitable access to cancer services by standardizing treatment protocols and referral pathways across both public and private facilities. This can help to reduce disparities in access to care and ensure that all patients receive timely and appropriate treatment [37,38]. The successful implementation of the NHI will require addressing workforce concerns, such as the disparity in pay and workload between the public and private sectors. The NHI can promote greater equity in compensation and benefits by establishing national standards for healthcare worker salaries and benefits [39]. Additionally, the NHI can support joint professional development opportunities for healthcare workers from both sectors to enhance their skills and knowledge and to promote a sense of shared purpose [39].

##### Address Financial Barriers to Care

This directly addresses Affordability, ensuring that financial constraints do not prevent patients from accessing life-saving radiotherapy. Patients in Gauteng face financial burdens related to transportation, accommodation, and meals. Financial assistance programs, such as transportation vouchers, subsidized lodging near treatment centers, and meal stipends, to reduce out-of-pocket expenses for patients, particularly those from low-income backgrounds, should be expanded [40,41,42,43]. Cost-sharing mechanisms, such as tiered pricing based on income or a sliding scale for co-payments should be implemented to make cancer treatment more affordable. The effectiveness of these programs should be evaluated through regular monitoring of patient financial burden, access to care, and treatment outcomes. Finally, increased government funding for cancer care to alleviate the financial strain on patients and families should be advocated [43].

##### Improve Patient Education and Support Services

To improve accommodation and acceptability of cancer care, comprehensive patient education and support services are essential. These services should empower individuals to seek timely and appropriate care, navigate the complexities of cancer treatment, and improve their overall well-being, and provide clear and accurate information about cancer prevention, early detection, treatment options (including potential side effects), and available supportive care resources [44]. This information should be culturally sensitive and accessible in multiple languages to cater to the diverse population [35,45].

By addressing these challenges and implementing these recommendations, policymakers can create a strong foundation for equitable and effective cancer care in Gauteng.

### 3.2. Pillars: Core Components of the Radiotherapy System

Building upon the foundation of Policy & Governance, this section delves into the core components that constitute a robust and functional radiotherapy system. These components, visualized as the pillars of the House framework are interconnected and essential for delivering equitable cancer care. They are service delivery, health workforce, information, medical products/technologies, financing, and leadership/governance. Strengthening these pillars is crucial for ensuring that radiotherapy services are available, accessible and of high quality [1,10]. However, in Gauteng, South Africa, public facilities, which serve most of the population, face significant challenges that can undermine these pillars. As highlighted in the mixed-methods assessment described earlier, public facilities often grapple with higher patient loads, limited equipment, and staffing constraints compared to private centers. These issues can lead to longer waiting times and impact treatment effectiveness. Addressing these specific challenges within the framework of the core components and the 5As of access is vital for improving the overall radiotherapy system.

#### 3.2.1. The Challenges in Public Facilities in Gauteng

Public facilities in Gauteng, which serve most of the population, face significant capacity constraints compared to private centers. These constraints manifest as the following:Higher patient loads: the sheer volume of patients seeking treatment at public facilities strains resources and contributes to longer waiting times, directly impacting the accessibility of care. As noted in the qualitative study, some patients reported waiting months for initial consultations.Limited radiotherapy equipment: the availability of functional radiotherapy machines is a major concern, affecting the availability of services. The mixed-methods assessment indicated that public facilities often have fewer machines per patient compared to private facilities. Equipment downtime further exacerbates this issue.Staffing shortages: a shortage of trained personnel, including radiation oncologists, medical physicists, and radiation therapists limits the capacity of public facilities to provide timely and effective treatment, reducing availability. This can also affect the acceptability of care if patients feel rushed or uncared for.

Resource allocation imbalance: the imbalance in resource allocation between the public and private sectors contributes to disparities in access and quality of care. This imbalance affects affordability for those who cannot afford private care and accessibility due to overburdened public facilities.

Equipment maintenance: public facilities often struggle with equipment maintenance due to budget constraints and bureaucratic processes, directly impacting the availability of radiotherapy services. As one healthcare professional noted, “We often have to wait months for repairs, which leads to treatment delays.”

Waiting times: the combination of these challenges results in longer waiting times for patients seeking radiotherapy treatment in public facilities, impacting the accessibility of care. These delays can lead to disease progression and reduced treatment effectiveness. Furthermore, long waiting times and inconvenient appointment scheduling can negatively affect accommodation, as the services are not organized to meet patient needs. Some patients expressed frustration with the lack of clear communication about treatment processes during the cancer care continuum, further impacting accommodation.

#### 3.2.2. The Proposed Solutions to Strengthen Each Pillar of the House Framework

To address the challenges outlined above and aligned with the “House” framework, the following solutions are proposed:

##### Increase Investment

Prioritize investment in specific radiotherapy equipment based on needs assessments of public facilities. For example, invest in modern linear accelerators with advanced imaging capabilities to improve treatment accuracy and reduce side effects [10,27]. Also, dedicate funds for regular maintenance and upgrades of existing equipment to ensure their continued functionality [28]. These investments directly address the challenges of limited equipment and infrastructure in public facilities.

The workforce investment should focus on radiation oncologists, medical physicists, radiation therapists, and oncology nurses. Offering competitive salaries and benefits can attract and retain qualified professionals, addressing the critical workforce shortage in the public sector [25,46]. Strategies such as scholarships or loan repayment programs for students pursuing careers in oncology can also be explored. Furthermore, investment should also be directed towards the acquisition and implementation of AI-powered tools for radiotherapy treatment planning [47,48,49]. AI can assist in various aspects of personalized medicine by using multiple data analytics algorithms [50]. This plan should be monitored and evaluated regularly to ensure that resources are being used effectively.

##### Streamline Processes

Implement specific strategies to streamline referral and treatment planning processes. To address the challenges of equitable access to radiotherapy within Gauteng, and in alignment with the goals of a broader national cancer control plan, specific strategies to streamline referral and treatment planning processes are crucial. Recognizing that NCCPs will provide overarching frameworks, this section details actionable steps for implementation at the local level. The public healthcare sector in South Africa faces a shortage of oncology healthcare staff, including medical physicists, radiotherapists, oncologists, and trained nurses. This shortage is exacerbated by the fact that 25% of registered oncologists are responsible for the care of more than 75% of the population. Prior to implementation, a process mapping exercise should be conducted to identify bottlenecks in the referral and treatment planning pathways. Based on the needs assessment to identify specific infrastructure gaps, specific interventions can be developed and implemented. The intervention should include investment in radiation oncologists, medical physicists, radiation therapists and oncology nurses, which will directly improve the ‘Health Workforce’ pillar. These specific interventions should be developed and implemented to address bottlenecks, such as electronic referral systems, standardized treatment protocols, and AI-driven scheduling tools [49,51,52]. Moreover, they should be aligned with the broader goals of a NCCP, such as improving efficiency, reducing disparities, and ensuring patient-centered care:Develop and implement standardized referral guidelines with clear criteria for radiotherapy eligibility.Establish a multidisciplinary tumor board with representatives from different specialties to facilitate communication and coordination of care.Implement AI algorithms to assist in treatment planning, potentially reducing the time required and optimizing treatment plans for individual patients. This aligns with improving the accommodation of patients.Employ AI-powered predictive analytics to optimize scheduling and resource allocation, reducing waiting times and improving service delivery efficiency.

##### Enhance Patient Support

A patient support center at each public radiotherapy facility should be established, while patient navigators should be recruited to provide individualized support to patients and their families. Patient education materials in multiple languages and formats should also be developed and disseminated. This improves affordability, acceptability, and accommodation, ensuring patients are well-informed, financially supported, and emotionally prepared for treatment [53].

Provide comprehensive patient support services, including information and education about radiotherapy treatment, financial assistance, and psychosocial support.Develop patient navigation programs to help patients navigate the complexities of the cancer care system.

### 3.3. Roof: Equitable Cancer Care & Improved Outcomes

The aim of this framework, represented by the roof of the ‘House,’ is to achieve equitable cancer care and improved outcomes for all patients in Gauteng, regardless of their socioeconomic status or geographic location. Equitable cancer care, in this context, means ensuring that all individuals have equal access to timely diagnosis, appropriate treatment, and patient-centered support, ultimately eliminating disparities in cancer outcomes across different population groups. Improved outcomes encompass increased survival rates, enhanced quality of life for patients and survivors, and reduced cancer-related morbidity.

Achieving this roof requires a comprehensive approach that addresses the 5As of access. Improving the availability of radiotherapy services, streamlining referral processes to enhance accessibility, providing financial assistance to ensure affordability, addressing cultural beliefs to promote acceptability, and organizing services in a patient-centered manner to improve accommodation are all essential steps.

Furthermore, strengthening the health system building blocks—including service delivery, health workforce, information systems, access to essential medicines and technologies, financing, and leadership/governance—is critical for creating a functional and equitable radiotherapy system. Innovative solutions such as AI-driven technologies can further optimize treatment and resource allocation. For example, AI-powered diagnostic tools can improve the accuracy and speed of cancer detection, while AI algorithms can optimize treatment planning for personalized therapy [48,49].

However, achieving equitable cancer care and improved outcomes is an ongoing process with inherent challenges. It requires sustained commitment, collaboration, and innovation to overcome systemic barriers and ensure that all cancer patients in Gauteng receive the best possible care [44].

The House framework emphasizes a holistic approach to improving radiotherapy access in Gauteng, with the roof representing the goal of equitable cancer care and improved outcomes. A key element of this framework is the focus on the 5As—Availability, Accessibility, Affordability, Accommodation, and Acceptability—which serve as concrete pathways toward achieving this goal. The subsequent discussion will elaborate on each of these 5As, highlighting their specific contributions to equitable cancer care:

The 5As as pathways to equitable outcomes: equitable cancer care and improved outcomes are achieved when the 5As are effectively addressed throughout the cancer care continuum:

Availability: Enough radiotherapy facilities, equipment, and trained personnel are essential to meet the needs of the population. There are disparities between public and private facilities, with public facilities often facing limitations in equipment and staffing.

Accessibility: Geographic and structural barriers to accessing radiotherapy services must be minimized. This includes addressing transportation challenges, streamlining referral processes, and establishing services in underserved areas.Affordability: Financial barriers, such as high treatment costs, transportation expenses, and lost income, must be addressed to ensure that all patients can afford radiotherapy treatment. The findings emphasize the financial burden on patients, with many reporting significant out-of-pocket expenses.Accommodation: Healthcare facilities should be organized and services delivered in a way that meets the needs and preferences of patients. This includes flexible appointment scheduling, shorter waiting times, clear communication about the treatment process, and culturally sensitive care. The patients expressed confusion about the treatment process and a desire for more information.Acceptability: Cultural beliefs and other factors that may influence patients’ willingness to seek and adhere to radiotherapy treatment must be addressed. This involves providing culturally sensitive care, addressing the stigma associated with cancer, and promoting trust in the healthcare system.

While the 5As offer a valuable checklist for identifying and addressing barriers to equitable cancer care, the House framework provides a roadmap for practical implementation. It emphasizes that sustainable improvements require not only addressing the 5As at the point of care but also strengthening the underlying policy and governance structure and core components of the health system. The Foundation and Pillars of the House framework facilitate the effective implementation of the 5As as follows:Foundation: Sound policies and governance structures are essential to ensure equitable resource allocation and effective cancer control planning. These policies should prioritize the 5As to create a supportive environment for cancer care.Pillars: Strengthening the core components of the healthcare system (service delivery, health workforce, information, medical products/technologies, financing, and leadership/governance) is crucial for improving the 5As and achieving equitable outcomes.Roof: By addressing the 5As and strengthening the foundation and pillars of the “House” approach, the goal of equitable cancer care and improved outcomes can be achieved for all patients, regardless of their socio-economic status or geographic location, as evidenced by the qualitative study of patient experiences.

Addressing the 5As and strengthening the Foundation and Pillars of the House framework are essential steps toward equitable cancer care. To ensure that these efforts translate into tangible improvements for patients, it is necessary to track key indicators related to access, treatment outcomes, and patient experiences. The following section outlines these metrics, providing a means of measuring our progress toward equitable outcomes:Monitor key indicators related to access, such as waiting times, travel distances, and out-of-pocket expenses.Track cancer incidence, mortality, and survival rates across different socio-economic and geographic groups.Regularly assess patient satisfaction and experiences to identify areas for improvement.

By focusing on the 5As and taking a holistic approach to cancer care, healthcare systems can make significant strides toward achieving equitable cancer care and improved outcomes for all patients.

## 4. Discussion

This study confirms persistent disparities in radiotherapy access between public and private facilities in Gauteng, with public facilities facing higher patient loads and resource constraints [9,51]. While these findings align with prior research on strained healthcare systems in South Africa and other LMICs this study provides further evidence that these disparities contribute to significant delays in treatment, negatively impacting patient outcomes and potentially exacerbating disease progression [52]. This imbalance in resource allocation between the public and private sectors highlights systemic issues within the Foundation: Policy & Governance component of the House framework, where policies fail to ensure equitable distribution of resources. The public healthcare sector in South Africa also faces a shortage of oncology healthcare staff, including medical physicists, radiotherapists, oncologists, and trained nurses. This shortage, exacerbated by the disproportionate distribution of oncologists, directly weakens the Pillars: Core Components of the Radiotherapy System, specifically the Health Workforce pillar, undermining the capacity to deliver timely and quality care. There is only 25% of registered oncologists responsible for the care of more than 75% of the population [1,5,6]. These delays underscore the urgent need for policy interventions aimed at optimizing resource allocation and strengthening healthcare infrastructure (reinforcing the Foundation and Pillars). Globally, similar disparities are observed, with LMICs often having significantly fewer radiotherapy units per capita compared to high-income countries [10]. This resource gap contributes to a substantial treatment gap, where a significant proportion of cancer patients in LMICs lack access to potentially life-saving radiotherapy [53]. Addressing this global inequity in radiotherapy access requires international and local efforts to ensure resources are available, ultimately striving towards the ‘Roof: Equitable Cancer Care & Improved Outcomes’ for all patients [54].

While this study provides valuable insights into radiotherapy access challenges within Gauteng, South Africa, it is important to acknowledge the limitations of generalizing these findings to other contexts due to the unique socio-economic and healthcare landscape of the region. Radiotherapy access and cancer care delivery challenges exist across diverse settings. Therefore, to provide a more comprehensive understanding of global radiotherapy access, we must consider findings from other regions, particularly LMICs, which often face critical shortages of radiotherapy machines and trained personnel.

In Canada, a high-income country with a universal healthcare system, geographic accessibility to radiotherapy centers significantly impacts cancer outcomes. This highlights a weakness in the Pillars: Core Components of the Radiotherapy System, specifically the Accessibility pillar, even in well-resourced settings. Chan et al. found that longer distances to treatment centers, particularly in northern regions, contribute to poorer cancer outcomes, independent of sociodemographic factors [55]. As the study points out, indigenous people are much more likely to live further away from a radiotherapy center in Canada. This finding underscores that disparities in access are not limited to LMICs and can exist within high-income nations due to geographical barriers, thereby hindering the achievement of the Roof: Equitable Cancer Care & Improved Outcomes for all Canadians. This situation also reflects potential shortcomings in the Foundation: Policy & Governance component, where policies may not adequately address geographical barriers to care and ensure equitable resource distribution across all regions [56].

Similarly, a national survey of cancer service providers in Australia identified significant gaps in meeting the bio-psycho-social and long-term care needs of cancer survivors [57]. This indicates a need to strengthen the Pillars: Core Components of the Radiotherapy System, specifically the Supportive Care pillar, to ensure holistic patient well-being. The study emphasized the need for improved coordination of services, particularly for complex patients and those from diverse communities, highlighting the importance of integrated care models. This also reflects a challenge in Foundation: Policy & Governance to effectively coordinate healthcare services and allocate resources to address diverse patient needs. Australia’s experience shows that even with regional radiotherapy centers, utilization rates in rural areas lag behind metropolitan areas [58]. According to Thompson et al., establishing regional centers alone is not enough; socio-economic and cultural factors also play an important role [58]. Therefore, equitable access, as envisioned by the Roof: Equitable Cancer Care & Improved Outcomes, is not solely dependent on geographical proximity but requires a comprehensive approach that considers socio-economic and cultural determinants, alongside robust supportive care and well-coordinated policies.

Furthermore, a systematic review of the global literature on distance travelled for radiotherapy revealed that the travel burden has detrimental consequences on patients’ mental health, participation in clinical trials, and treatment outcomes [59]. This directly impacts the Roof: Equitable Cancer Care & Improved Outcomes, as these burdens disproportionately affect vulnerable populations. The review also noted that rural and socioeconomically disadvantaged patients are disproportionately affected by travel burdens, further exacerbating inequities in cancer care. These inequities exist in both high-income countries such as Canada and Australia as well as LMICs, highlighting a systemic issue in the Foundation: Policy & Governance that fails to address the needs of these populations. The systematic review represents the most comprehensive analysis of the distance travelled for RT globally, revealing that these geographical disparities hinder the ‘Accessibility’ pillar within the Pillars: Core Components of the Radiotherapy System [59].

Global studies have consistently shown a significant disparity in access to radiotherapy services, with LMICs facing a critical shortage of radiotherapy machines and trained personnel [4]. Addressing these shortages necessitates a focus on the House framework. The shortage of machines and personnel represents a clear failing of the Pillars: Core Components of the Radiotherapy System, particularly regarding the components of Infrastructure and Health Workforce. The IAEA recommendation of 250,000 population per megavoltage machine highlights the disparities in access to treatment. International collaboration, as highlighted by Jackman et al., directly addresses the Foundation: Policy & Governance, where international agreements and national policies should prioritize resource allocation and workforce development, strategic investment in infrastructure and technology is essential [60]. Continued emphasis on the importance of international collaboration involving academia, government, and industry to enhance global accessibility and equality in radiotherapy is critical. Over half of patients worldwide lack access to this treatment, severely hindering the achievement of the Roof: Equitable Cancer Care & Improved Outcomes [53]. The WHO states that the highest attainable standard of health is a fundamental right of every human being [60].

To address the challenges in LMICs, various strategies have been proposed, reflecting a comprehensive approach aligned with the House framework. Focusing on the Core Components of the Radiotherapy System, the development of improved Cobalt-60 units capable of modern dose delivery and the manufacturing of simple yet robust linear accelerators suitable for locations with challenging infrastructure directly address the need for accessible and appropriate technology [26,61]. The implementation of telemedicine interventions further supports this pillar by expanding access to expertise and care. Strengthening the Foundation: Policy & Governance, collaborative initiatives like the Africa Radiation Oncology Network, an IAEA telemedicine pilot project, facilitate online case discussions and knowledge sharing among radiation oncologists in different African countries, extending specialist expertise to underserved areas [61,62]. Another promising strategy, the adoption of hypofractionated radiotherapy, which shortens the overall treatment duration by delivering higher doses per fraction, has the potential to increase patient throughput and reduce costs, making radiotherapy more accessible in LMICs [10,63,64]. These strategies contribute to the Roof: Equitable Cancer Care & Improved Outcomes, ensuring more patients receive timely treatment. For instance, Washington University in St. Louis collaborated with Guatemala to enhance radiation therapy delivery, bridging a significant healthcare gap. Countries like Rwanda have experimented with task shifting and retraining of general practitioners with reported success on a small scale [65,66].

While challenges in infrastructure, workforce, and financial resources present significant obstacles to radiotherapy access in Gauteng, the House framework emphasizes that equitable cancer care and improved outcomes (the Roof) can only be achieved through a strong foundation of policy and governance and well-supported core components (the Pillars). To address these challenges, public–private partnerships offer a potential solution by leveraging private sector resources and expertise to improve radiotherapy infrastructure and service delivery [60,66]. As Jackman et al. highlighted, encouraging public–private partnerships is crucial for addressing disparities in radiotherapy access [60]. For example, build-operate-transfer schemes have proven feasible for investment in modern technology [10,67,68]. Moreover, the IAEA actively fosters collaborative partnerships between the public and private sectors to enhance global accessibility and equality in radiotherapy. Furthermore, staff shortages, driven by remuneration disparities and challenging working conditions, emerge as critical barriers to providing timely and effective radiotherapy care [66]. To achieve equitable cancer care, policymakers must prioritize cancer control, strengthen regulatory frameworks, and allocate sufficient resources to support the expansion and sustainability of radiotherapy services. Therefore, a comprehensive re-evaluation of the existing policy landscape is warranted.

### Adaptability and Generalizability

One limitation of this study is its focus on Gauteng province, which may limit the generalizability of the findings to other regions with different healthcare systems. Despite these limitations, the House framework offers a valuable foundation for understanding and tackling obstacles to cancer care in other resource-constrained environments. Its core principles, drawing from the WHO Health Systems framework and the 5As of access, allowing for flexible adaptation. Successfully implementing the framework in new settings requires a thorough and realistic needs assessment to identify specific infrastructure gaps, as highlighted in our analysis of Gauteng. Furthermore, our detailed methodology, encompassing a systematic review, quantitative analysis, exploration of structural quality indicators, and a qualitative study of patient experiences, provides a transparent and replicable roadmap for other researchers. Strategic investment in radiotherapy infrastructure and technology is essential to meet the growing demand for cancer treatment in Gauteng, particularly through upgrading imaging technology and standardizing equipment. These are key components for consideration in any LMIC.

## 5. Conclusions

This study successfully developed the House framework for improving radiotherapy access in Gauteng, South Africa. This research underscores the critical need for a multifaceted approach to address the challenges in radiotherapy access within Gauteng, advocating for the implementation of strategies that are both evidence-based and context-specific. By addressing these challenges, we can improve equitable access to radiotherapy services and ultimately improve cancer outcomes in Gauteng.

## Figures and Tables

**Figure 1 ijerph-22-01071-f001:**
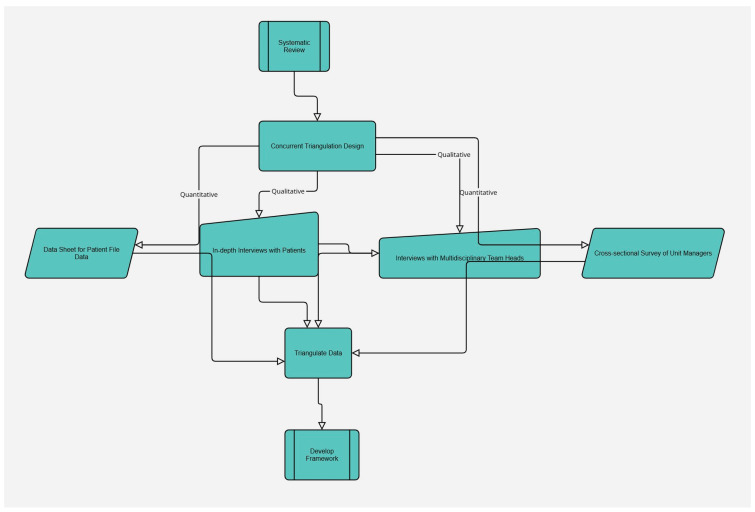
Adapted Methods.

**Figure 2 ijerph-22-01071-f002:**
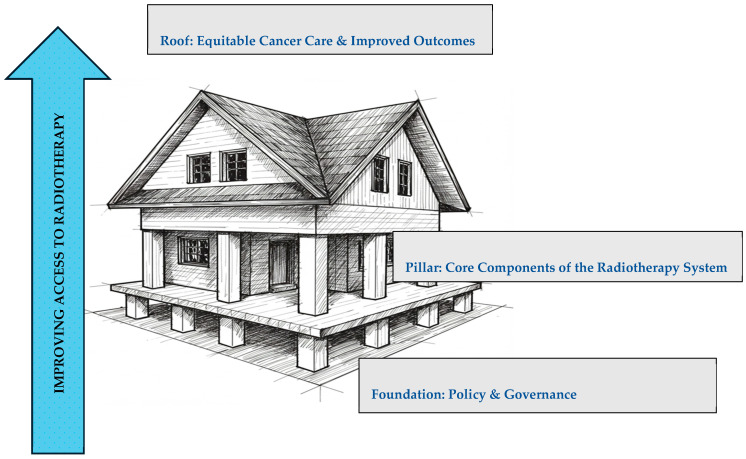
The House framework.

**Table 1 ijerph-22-01071-t001:** Themes from all the phases of the study.

Phase of Project	Study Focus	Main Themes	Supporting Citation
Systematic Review	Barriers to Radiotherapy Access in Sub-Saharan Africa	Health System Barriers, Patient Socio-demographic Barriers, Provider Factors	[13]
Quantitative Analysis	Time Intervals from Diagnosis to Treatment	Delays in Diagnosis, Delays in Treatment Initiation, Variations in Time Intervals	[14]
Mixed Methods Assessment	Structural Quality Indicators of Radiotherapy Services	Limited Radiotherapy Facilities, Staff Shortages, Equipment Challenges	[15]
Qualitative Study	Patient Experiences of Radiotherapy Access	Financial Burden, Lack of Information, Transportation Challenges, Psycho-social factor	[unpublished patient experience]
Framework Development	Strategies to Improve Radiotherapy Access in Gauteng	Infrastructure Development, Workforce Training and Retention, Task Shifting, International Collaboration, Financial Investment	

## Data Availability

All data in this study were provided in the main manuscript.

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
