# Peer review of "Improving Access to Radiotherapy in Gauteng: A Framework for Equitable Cancer Care"

_ijerph, 2025, doi:10.3390/ijerph22071071_

Round 1
Reviewer 1 Report
Comments and Suggestions for Authors
This paper describes a mixed methods design to improve access to radiotherapy services in Gauteng, South Africa. the project comprises of five phases - each for addressing specific aspects of radiotherapy access in this area. viz. systematic review, quantitative analysis of key time intervals, exploring structrual quality indicators, qualitative (patient experiences phase), policy development. House framework represents integrated framework developed for this purpose. it is well described study however following issues need to be addressed -
para in discussion from line 485-497 should have some relevant references.
Full form of GCCC is missing.
including a fig (with a flowchart) to describe adopted method is recommended.
more studies in other parts of the world should be cited in discussion to make it more comprehensive. (PMID: 35960905, 29179701 etc.)
no. of study participants could be mentioned in study design and their data supplementary data.
Reviewer 2 Report
Comments and Suggestions for Authors
Was the development of the 'House' Framework logically derived from the data across the five study phases?
Is the integration of WHO’s health system building blocks with Penchansky & Thomas’s '5As' of access methodologically sound?
Did the concurrent mixed-methods design ensure comprehensive data triangulation?
Were data collection and analysis methods clearly described and appropriate?
Were standard credibility strategies (e.g., member checking, peer debriefing, inter-coder reliability) employed effectively?
Was the research ethically approved and compliant with university and national ethics guidelines?
Was informed consent obtained and reported transparently?
Can other researchers replicate this framework or apply it to other regions?
Are terms, references, and models used in a consistent and comprehensible manner?
Does the study present novel contributions to improving radiotherapy access in low-resource settings?
Is the ‘House Framework’ applicable beyond the South African context?
Comments on the Quality of English Language
Need to improve
Reviewer 3 Report
Comments and Suggestions for Authors
The paper is well-written and easy to follow. With large disparities between public and private sector it would be interesting to see how to facilitate greater monetary investment into the public sector to improve equipment, personnel needs, etc. In addition, it would be interesting to see how a collaborative effort with private sector could be brought about to diminish disparities between it and public sector.
Very minor editorial suggestions (please see attached document).
Additional comments:
Abstract:
The authors mention that the manuscript will discuss hypofractionation but there is no other mention of this approach to cancer therapy other than a reference in the citation list.
Introduction:
No additional comments.
Materials and Methods:
The bullet for that bullet point should be on line 130 To keep the materials and methods section focused on the current phase of the larger project, it might be helpful to add the summary language from sections 2.3.1, 2.3.2, 2.3.3, and 2.3.4 to the appropriate sections in 2.3.5 (you wouldn't need all of the individual subsections). This would shorten the materials and methods section a bit and also put the findings from each of the previous parts of the project together with the summary material in sections 2.3.1-.4.
Line 151 should be to left margin and not indented with bullet points.
The following paragraph starting at line 159 is redundant with previous bullets (lines 152-158) so recommend removing either the bulleted list or the paragraph. This reviewer feels like the paragraph is more informative than the bulleted list, so would recommend removing the bulleted list.
Results:
Check formatting of section heading as it is inconsistent with that of other main section headings.
If a National Cancer Control Plan is developed, would there be subsections by cancer type that would apply to both the public and private sector? Would this be an establishment of standard of care for each cancer type that would apply to all sectors?
Section 3.1.3.2 - How would you argue for allocation of funds for increased investment? Would funds be reallocated from somewhere else in the budget or where would additional funds come from?
Section 3.1.3.3 - How would you convince the private sector to collaborate? If the private sector is for profit, would there be any risk to their profit margin through collaboration? If health care workers left the public sector to go to the private sector due to poor pay or workload, etc., what steps would be taken to make such collaboration palatable to workers from both sectors?
Section 3.2.2.1 - Is the investment described in this section the same as or related to that described in 3.1.3.2?
Section 3.2.2.2 - Would streamlined processes be part of or related to a National Cancer Control plan or standard of care that would apply to both public and private sectors?
AI algorithms will need extensive testing and validation to be reliably safe to use in treatment planning.
Section 3.3 Roof - check formatting so that it is consistent with other section headings Line 430 - "The subsequent section . . . " This line is unnecessary as the language that follows is still part of the same section.
Discussion:
The Results section of the manuscript is long and has many elements of discussion in it. To help pull everything together, would it be helpful to list out what steps in order the authors recommend to achieve the stated goal of equitable radiotherapy?

Round 2
Reviewer 2 Report
Comments and Suggestions for Authors
Accept at present form
Comments on the Quality of English Language
Ok